# Improving Lodging Resistance: Using Wheat and Rice as Classical Examples

**DOI:** 10.3390/ijms20174211

**Published:** 2019-08-28

**Authors:** Liaqat Shah, Muhammad Yahya, Syed Mehar Ali Shah, Muhammad Nadeem, Ahmad Ali, Asif Ali, Jing Wang, Muhammad Waheed Riaz, Shamsur Rehman, Weixun Wu, Riaz Muhammad Khan, Adil Abbas, Aamir Riaz, Galal Bakr Anis, Hongqi Si, Haiyang Jiang, Chuanxi Ma

**Affiliations:** 1School of Agronomy, Anhui Agricultural University, Hefei 230036, China; 2Key Laboratory of Wheat Biology and Genetic Improvement on South Yellow & Huai River Valley, Ministry of Agriculture, Anhui Agricultural University, Hefei 230036, China; 3National Engineering Laboratory of Crop Stress Resistance Breeding, Anhui Agricultural University, Hefei 230036, China; 4Department of Plant Breeding and Genetics, University of Agriculture Peshawar, Peshawar 57000, Pakistan; 5School of Life Sciences, Anhui Agricultural University, Hefei 230036, China; 6State Key Laboratory for Rice Biology, China National Rice Research Institute, 359#, Tiyuchang Road, Hangzhou 310006, China; 7Rice Research and Training Center, Field Crops Research Institute, Agriculture Research Center, Kafrelsheikh 33717, Egypt

**Keywords:** lodging, morphological management, plant growth regulators, resistance genes, agronomical management

## Abstract

One of the most chronic constraints to crop production is the grain yield reduction near the crop harvest stage by lodging worldwide. This is more prevalent in cereal crops, particularly in wheat and rice. Major factors associated with lodging involve morphological and anatomical traits along with the chemical composition of the stem. These traits have built up the remarkable relationship in wheat and rice genotypes either prone to lodging or displaying lodging resistance. In this review, we have made a comparison of our conceptual perceptions with foregoing published reports and proposed the fundamental controlling techniques that could be practiced to control the devastating effects of lodging stress. The management of lodging stress is, however, reliant on chemical, agronomical, and genetic factors that are reducing the risk of lodging threat in wheat and rice. But, still, there are many questions remain to be answered to elucidate the complex lodging phenomenon, so agronomists, breeders, physiologists, and molecular biologists require further investigation to address this challenging problem.

## 1. Introduction

The primary causes of plant lodging including: legion, increased nitrogen levels, over plant population, soil density, diseases, natural disasters such as storm damage, sowing date, and seed type, are all mainly contributing factors to lodging in cereal crops. Lodging refers to stem breaking type and stem bending type (stem lodging) [1] or root lodging (anchorage failure) [2] of the plants, and is one of the most concerning problems faced by the farmers worldwide [3]. Generally, the possibility of lodging occurs when the plant weight (upper parts of plants) increased by the interception of rainfall. When the lower stem parts are weakened by disease attack or by overdose application of nitrogenous fertilizer, or when the shearing cohesive bond strength of the soil particles around the root system is almost completely deteriorated by rainfall. Lodging severely affects grain production of the major cereal crops; particularly wheat and rice also have several other indirect knock-on effects such as crop harvest at a slower pace, reduction in grain quality, and drying costs [4,5,6]. These factors are considered among some of the major constraints in reducing crop productivity globally.

In the United Kingdom, lodging in winter wheat accounted for about $80 million annual losses to the farming industry [7]. Reduction in the grain yield up to 80% has been estimated in a study focusing on the impact of lodging on crop yield using both natural and artificially-induced lodging modes [8]. Niu et al. [9] reported that robust storms-continuous rainfalls and heavy rainfalls were the major factors for lodging to reduce wheat production in China. Overall, robust storms-continuous rainfalls accounted for 73% lodging while heavy rainfalls and robust storm contributed to about 19% and 8% lodging, respectively. In Egypt, the reduction in wheat grain production by 7.2% and 19.9% was observed by lodging at 275 kg N ha^−1^ and 225 kg N ha^−1^, in comparison with 175 kg N ha^−1^ and 150 kg N ha^−1^ applications, respectively [10]. Artificially-induced lodging in wheat at the ear emergence, soft dough, hard dough, and at milk stages decreased the grain yield by 31%, 20%, 12%, and 25%, respectively [11]. Generally, in wheat, grain yield reduction is associated with the crop growth stage at which the lodging occurs, like the onset of stem elongation and flowering. During these stages, the wheat crop is highly vulnerable to damage by wind and frost [12].

Furthermore, Berry and Spink [4] reported that crops falling within an angle of 45° may result in the 18% decline of grain production. Similarly, plants lodging at an angle of 80° from the perpendicular position during the flowering phase could reduce the significant proportion of grain yield. The reduction in the grain yield varies from 8–34% [11], up to 54% [4], and 43–61% [8], had revealed from different studies. CIMMYT (International Maize and Wheat Improvement Center) investigators have also observed a similar proportion of wheat area affected with lodging in North West Mexico [13].

Lodging is associated with height reduction due to a bending of the shoot from the vertical position. This is most prevalent in the canopies of modern irrigated and deep-watered rice growing in the fields under tropical conditions and is accompanied by yield reductions up to 2 t ha^−1^ [14]. They also reported that lodged plants during stem elongation and grain-filling stages had a 40% reduction in yield by using a semi-dwarf rice cultivar. It has been observed that lodging during the grain filling stage occurring at 25–90° angle from the perpendicular could reduce grain yield which varies from 20–61% in wheat [4].

Weng et al. [15] described that stem lodging in rice significantly disturbed the photosynthetic capability of the canopy by affecting the grain filling stage. As a result, reduction in grain yield and quality, higher crop harvest cost, and severe inhibition of the balanced yield in rice occurred as reported by Haghdoost et al. [16]. For example, lodging negatively affected both rice grain yield and quality [17] on account of 60–80% by reducing rice canopy photosynthesis [18]. Tall rice genotypes either artificially induced to lodge or allowed to lodge showed a 35% reduction of 2 t ha^–1^ in grain yield [19]. Grain yield decreased up to 80% in the wet season and up to 50% in the dry season due to lodging in the thin stemmed genotypes, despite the abundant nitrogen application [20,21,22]. Moreover, lodging stress posed difficulties in machine harvesting and hampered the working efficiency by up to 25% [23].

In wheat and rice, the traits which are most commonly associated with lodging resistance are plant height, culm diameter and thickness, strength of upper and lower internodes, thickness of stem wall, lignin and cellulose accumulation in the stem wall, and spike weight (Figure 1) [24]. This review article focused on the recent knowledge about the plant traits which primarily play a crucial role in improving lodging resistance in wheat and rice. Various approaches, like genetics, agronomic, and chemical that reduce lodging in wheat and rice are also discussed. Moreover, we analyzed the basis of lodging, and enlighten the fact that any cultivar susceptibility to lodging depends on three main factors: (i) what are the exact size and dynamics of the forces to which it is exposed to? [25], (ii) the correlation of stem bending strength to lodging resistance [26], and (iii) the anchorage power of the root system.

## 2. Morphological Traits and their Responses under Lodging Stress

### 2.1. Plant Height and Inter-Nodal Length

Plant height plays a pivotal role in the lodging resistance of cereal crops [15,27,28]. Plant height is strongly associated with lodging resistance (Table 1) at all developmental stages in cereals [29,30]. Thus, their strong and positive association and the subsequent relationship of plant height with yield and yield-related components have previously been reported by Yu et al. [31]. Genotypes with tall height are highly prone to lodging, in contrast with the genotypes which have shorter plant height as they have the potential to withstand under lodging pressure [32,33].

At the flowering time and milky phase of grain formation in wheat [34], the weight increment of the spikes occurs. It increases the center of gravity of height and the continuous translocation of material deposited in the stem to spikes reduces the stem bending strength [30,35], which in turn negatively affects the plant’s storm resistance capacity.

Another factor is desirable internode length in rice and wheat, which enhances the resistance to lodging, supporting the fact that plant height is not only the single key element which is associated with lodging resistance [5,36,37] but also explains the enhancement in stem strength of the basal portion of the culm internode: is one of the key factors to develop resistance against lodging. Moreover, they reported a positive relationship of internodes number, plant height, and inter-nodal length with the lodging score, which depicts the role of these plant attributes for lodging resistance in wheat.

Studies have revealed that resistance to lodging is entirely dependent on and influenced by internodal length, plant height, and stem bending strength [38]. Reduction in plant height develops tolerance to lodging on the account of comparatively low center of gravity and reduction of the above-ground load of plant on the lower stem in rice [39,40,41], while semi-dwarf varieties in rice decreased their internode length, which then improved their tolerance to lodging. The short length of primary internodes can support to construct an ideal culm structure for resistance to lodging whereas the reduction in the length of the upper internode could have an adverse impact on panicle exertion [42,43,44]. A negative relationship of the basal internode length with lodging score has been observed, which suggests that the longer internodes at base could result in a higher lodging score in wheat [45] and rice [46].

### 2.2. Culm Wall Thickness, Culm Diameter, and Panicle Weight

Culm wall thickness and culm diameter confer resistance to lodging [39]. Li et al. [33] reported that plant culm diameter was generally greater at the first internode but reduced progressively in the upper direction of the plant portion. It was also observed that greater culm diameter strongly associated with the culm wall thickness which has an integral element to improve resistance to lodging in wheat and rice [36,39,44,47]. Likewise, culm diameter and stem strength in wheat and rice are correlated with length of the last internode from the base of the plant, leaf sheath length, and the cross-sectional area of stem [11]. Heavier and thicker culms also confer resistance to lodging in cereal crops particularly in wheat and rice [48,49].

Culm thickness is substantially associated with the resistance conferred by culm diameter and the lowest three internodes of the rice plant [50]. Moreover, the length of elongated internode of basal stem, culm wall thickness and culm diameter greatly influence the lodging resistance capacity in rice [51,52,53].

Cui et al. [54] reported that the expression of the culm diameter of the second internode resulted from dominant and additive gene effects. A significant variation for culm wall thickness in the second internode region among lodging-resistant and susceptible wheat genotypes (0.75 mm and 0.69 mm, respectively) have been observed [27]. Similar results were reported in a study of four common wheat cultivars, where a significant association was observed for lodging resistance with culm wall thickness (R = 0.972) and heaviness of the lowest three internodes (R = 0.986) [37] (Table 1). Subsequently, it has also been reported that lodging-resistant wheat genotypes have higher culm wall thickness and culm diameter of the lower internodes [55,56,57].

Under lodging stress, reduction in grain yield was linked to panicle weight and grain weight per panicle, as described in the past experiments [58]. Days to panicle and maturity were negatively correlated with lodging, while culm diameter, culm length, panicle weight, and panicle length were significantly and positively correlated (Table 1) with resistance to lodging in rice [59]. In the context of lodging index, it has been reported to correlate positively with the length of the internode, plant height, and gravity height, but remarkably negatively associated with both cellulose and lignin contents in wheat plant [60]. Stem bending movements in the direction of soil was the primary causal factor of lodging which occurred when rice panicle [61] or wheat spike [8,62] gained high weight at the period of maturation or due to some environmental factors like wind and rain. Nevertheless, the positive associations of panicle neck thickness (Table 1) with the number of panicle neck vascular bundles were reported by Xu et al. [63] and panicle neck angle lower than 40° was considered as an erect-type panicle [64]. In addition, lodging incidences were reduced by shorter peduncle with reduced center of gravity causing an increase in panicle erectness (Figure 2) [64,65].

### 2.3. Root Lodging

Root system architecture (RSA) primarily plays important roles in growth and development, anchoring the plant into its growth substrate, facilitating water and nutrient uptake from the soil. While, it is also considered crucial to combat under extreme environmental signals such as biotic and abiotic stresses, including root lodging [66]. Basically, plants have two main root systems: a taproot system and a fibrous root system. In case of wheat and rice, as they both are monocot species so contain fibrous root system [67]. It consists of a dense mass of adventitious roots (also called crown roots in cereals) that arise from the stem, which are distinct from the primary root, lateral roots (LRs), and root hairs [68]. Because the primary roots (embryonic root) dies within the growth age of monocots, whereas the adventitious roots are the main root tissues in the fibrous root system of monocot plants.

Root traits which are associated with root system architecture significantly contribute to overcome abiotic constraints and are critical to maintain the structural and functional properties of roots, and thus considered first order targets in the breeding programs. Root characteristics, such as deep root system, increased root density in subsoil, increased root hair length and density, and/or xylem diameters, may contribute to enhance the lodging resistance [69,70]. For example, rice varieties resistant to lodging developed more roots in deeper soil layers than lodging-susceptible varieties, while contribution of unit root weight to lodging tolerance was higher in deeper than in shallower soil layers. Their work confirmed that, in rice, a greater ability to form roots with a higher bulk density in subsoil was one of the most important characteristics for root lodging tolerance [71]. Moreover, researchers in References [72,73] studied the mechanism of roots and root systems, and explicitly explained the implication of roots, root system, and soil factors in lodging phenomenon in winter wheat. They reported that, lodging resistance was not related to the strength and stiffness of the stems but rather was dependent on a cone of rigid coronal roots which emerge from around the stem base. Despite this, one theoretical model of anchorage suggested that lodging resistance should be dependent on the diameter of the root–soil cone, coronal root bending strength, and soil shear strength. Aerenchyma formation in roots is another key element which is considered to be linked to establish stronger and deeper roots to support lodging tolerance. In wheat, adventitious roots formation and shoots growth have positive association even under hypoxia condition [74]. In rice, formation of primary aerenchyma provides a favorable condition to grow plants well by improving root efficiency in flooded paddy conditions and enhances the tolerance to lodging; thereby they are considered to play a vital role in lodging indirectly [75].

Furthermore, major attentions have also been focused on the root system and the associated mechanics of root anchorage, as most of the cereal crops, particularly wheat and rice, are more prone to root lodging in comparison to stem lodging [47]. Root lodging occurs when the underground roots grow straight while intact culms lean from the crown owing to heavier head/inflorescence and wet soil [34]. Likewise, Pinthus [25] reported that lodging is the result of either tender culms or weakly anchored roots which could be a consequence of genetic or environmental factors such as insects or disease invasions. Robust root anchorage mechanisms of any plant are reliant on two characteristics of the root system: the angle of spread of the basal coronal roots and the bending strength [34]. They also explained that the force required for lodging plants enhanced linearly with deflection before leveling slightly at an angle greater than 30° in wheat.

Several studies have reported the association between culm strength and vigorous root anchorage in the upper layer of soil in wheat [34,76]. Phenotypic differences between the root system of highly lodging-resistant and lodging-vulnerable cultivars are quite obvious. For instance, plants that were standing on bundles of flexible vertical roots and stem had no penetration into soil at great extent thereby, have shallow crown depth, which then decreased anchorage strength of wheat plants [77]. In wheat, a positive correlation has been observed between the culm lodging resistance and the spread of the coronal roots [78]: expressed as the angle from the vertical points at which these flexible roots penetrate into the soil [77]. This association seems to be having a distinct impact as it was observed in those cultivars which were displaying similarities to other lodging-resistant characters related to roots and stem [25]. In addition, the root lodging resistance depends on the diameter of the root-soil cone, the bending strength of the crown and the shear strength of the soil. It was reported that the root system of wheat was weaker than the shoot system in wet soil [72]. Sterling et al. [79] also found that root lodging may occur if the maximum bending moment at the base exceeds soil strength, which is the function of soil parameters and soil moisture contents.

Globally, other scientists have implicated the root architecture system and its failure as an accountable element in rice lodging. Although rice plant has relatively shallow rooting system [70], which penetrates perpendicularly into the soil, but spreading of roots has been considered an important factor in rice cultivation. For example, high yielding rice populations often had a large number of downward growing nodal roots, which generally penetrate and develop well in relatively deeper layers of soil [80].

In contrast, both wheat and rice have a different root architecture system. Izumi et al. [81] explained the branching pattern in rice roots and this is more likely to be herringbone type. Though this does not imply a particular specific anchorage system in rice plant, the time-course varies in the root growth compared to the branching system, which may affect the resistance or lead to root lodging. Besides the growth angle of nodal roots, the size of nodal roots is also considered as a fundamental element in determining the spatial distribution of the plant root system in soil, which was observed to be involved in crop yield and under lodging resistance [82]. Ogata and Matsue [83] demonstrated the clear image of the rice root system’s impact and its nature where they found that the crown root’s thickness at different intervals after sowing is a prominent decisive aspect in lodging resistance in direct-sown rice. Therefore, this is crucial to have a comprehensive and thoughtful understanding of mechanics in root anchorage and lodging resistance in rice. It is expected to improve the resistance to root lodging under a high ratio of carbon dioxide (CO_2_) if we have a great proportion of root biomass per tiller [84].

The contribution of unit root weight to enhance lodging resistance was extremely higher in deeper soil in comparison to the shallow soil layer [70]. Thus, their finding affirmed that, in rice, a greater tendency to develop roots under higher bulk density in subsoil has been acknowledged as one of the most essential characteristics for tolerance to root lodging. The rice plant roots anchorage strength and increments towards the lodging resistance were particularly determined at seedling and maturing stages [85]. At the 7th and 8th leaf stages, the diameter of the crown roots was observed to be thicker in lodging-tolerant rice cultivars in comparison with susceptible cultivars. Similar results of a positive association of crown diameter with enhanced lodging resistance have been observed at maturity stages of the plant [51].

### 2.4. Carbohydrates and Role of Lignin Biosynthesis

Lignin and cellulose contents are the major components of the cell wall and their components are essential for plant vigor, and against biotic and abiotic stresses including plant lodging [86]. A high concentration of lignin in vascular bundles can enhance cell wall strength and improves the physical strength of plant stalk. The total lignin contents of basal second internode of wheat and rice were significantly associated with the breaking stability and elasticity of stems [39,57]. Lignin accumulation in higher amounts enhances the physical stability of culm internode in wheat [5]. During secondary cell wall formation, lignin is accumulated in the carbohydrate matrix of the cell wall, making the entire plant body robust, enabling the plant to grow upwards [87,88]. Structural carbohydrates and lignin concentrations in the cell wall of the lower internodes were not consistently associated with lodging in wheat cultivars [89]. It has been reported that the arrangement and interaction of the different structural carbohydrates and lignin in the culm cell wall could be more important in lodging resistance in wheat. Culm and secondary cell wall strength in mutant rice genotypes [90] and wheat [37] were due to the accumulation of cellulose, lignin, and hemicellulose in a significant amount. Lignin and cellulose contents in the cell wall also enhance lodging resistance in rice [15,51].

The association of the lodging resistance with higher concentrations of lignin, pectin, cellulose, and protein in the plant stem has been reported in several studies [37]. The amount of hemicellulose and lignin of culms in susceptible varieties have been observed low, as compared to that of lodging-resistant wheat varieties [47,86]. Enzymatic activities of Tyrosine ammonia-lyase (TAL), phenylalanine ammonia lyase (PAL), and peroxidase (POD) enzymes were significantly associated with the lignin build-up and accumulation in wheat culms [91,92]. Lodging resistance in wheat, thus, could be improved through the lignin accumulation in higher amounts with the help of the enhanced activities of the enzymes involved for lignin accumulation. Here, we discussed and enlightened the some information about lignin biosynthesis, its regulatory mechanism, and pivotal role to prevent or increase the sensitivity to lodging.

Generally, the formation of lignin in plants is accomplished by an oxidative coupling reaction of three monolignols which are considered as building blocks i.e., coniferyl, sinapyl, and p-coumaryl alcohols [93] as illustrated in Figure 3. These monolignols are the synthesized products of phenylalanine through the general phenylpropanoid and by means of monolignol-specific pathways. After the succeeding dehydrogenative polymerization reactions which then produce the three units, namely: guaiacyl (G), syringyl (S), and hydroxyphenyl (H); they finally make a complex and three-dimensional structure of lignin polymer [93].

Some previously conducted experiments had characterized the expression pattern of wheat homologs of lignin biosynthetic genes. Studies showed that transcripts of these homologs were highly expressed in the stem tissue in comparison to other plant tissues, such as in leaf sheath and leaf blade. A significant correlation among *PAL6, C4H, 4CL1, C3H1, CCR2, F5H1*, and *F5H2* enzymes expression and lignin content was observed [94]. Moreover, *CCR1* and *CAD1* transcripts were abundantly present in the stem along with the greater activity of the corresponding enzymes than that of measured quantity in other tissues, whereas *COMT1* was expressed constitutively throughout the stem, leaf, and root tissues [95,96,97]. It has also been observed in lodging-tolerant wheat cultivars that the varying extent to resistance for lodging is considerably dependent on the amount of transcript abundance (i.e., *CCR1, COMT1*, and *CAD1* genes) and on the contribution of the corresponding enzymes higher activities in the stem following at the heading stage, while these elements also showed close association with lignin contents and with the mechanical strength of the stem [96]. Despite the undeniable key role of lignin in lodging, until now, the lignin biosynthesis pathway has not been elucidated widely in the wheat plant.

In the recent studies of Tran-Nguyen et al. [92], they examined the lignin biosynthesis in wheat cultivars and concluded the strong, close association between lodging resistance with lignin contents and the expression of 4-coumarate: CoA ligase1 (*4CL1*), cinnamoyl-CoA reductase2 (*CCR2*), p-coumarate 3-hydroxylase1 (*C3H1*), ferulate 5-hydroxylase2 (*F5H2*), and caffeic acid O-methyltransferase2 (*COMT2*) in the plant internode. They revealed that these genes were greatly expressed in the wheat tissues, indicating the significance of these genes in the intervening lignin accumulation in wheat culm. However, the underlying molecular mechanisms for lignin synthesis in wheat tissues are poorly understood [92].

On the other hand, some factors including cellular signaling molecules such as plant hormones, biotic, and abiotic stresses [98,99,100] play an integral role in the up regulation of lignin biosynthesis or may cause hindrance in its synthesis. Auxin and cytokinin, for instance, prompt the expression of genes related to lignin biosynthetic, peroxidase (Prx) in Zinnia elegans, and secondary growth/lignification [101]. In Arabidopsis, auxin accumulation was induced by hyper-gravity that led to a high expression level of selected lignin biosynthetic genes, and sequentially led to lignification in the inflorescence stem [102]. Previous experiments also showed that salicylic acid (SA) level is inversely linked to lignin contents in plants where lignin contents decreased by down regulation of specific lignin biosynthetic genes, and salicylic acid accelerated growth suppression in those plants. Nevertheless, genetic reduction of SA level was found to restore growth but not lignin contents [103].

In wheat, the relationship between lignin level and plant hormones that are associated with regulation of lignin biosynthesis was measured by the amounts of indole-3-acetic acid (IAA), isopentenyl adenosine (IPA), t-zeatin, and salicylic acid (SA) in two cultivars (cv. Harvest and cv. Kane) in the internode tissues. Results depicted that cv. Harvest was found to be more tolerant to lodging by containing the high intensity of IPA, t-zeatin, and salicylic acid in the internode than that observed in cv. Kane, whilst the IAA contents showed no differences [92]. In addition to cellular signaling, the environmental factors are also another contributing factor in lignin synthesis and deposition, for example, waterlogging, which decreased lignin content and led to a significant reduction in IPA and t-zeatin levels [92]. This may cause a reduction in leaf photosynthetic capacity [104,105] and in the accumulation of soluble carbohydrate in plant biomass [106]. Moreover, it leads towards the channeling of more phosphoenol pyruvate to glycolysis by triggering increased demand for soluble sugars, especially of glucose instead of shikimate pathway through which phenylalanine synthesis occurs, a crucial lignin precursor [107,108,109]. These modifications have vital consequences on the level or configuration of structural carbohydrate polymers, for instance on cellulose, hemicellulose, and lignin [110,111].

Further to this, Ma et al. [96] designed a research experiment to explore the Cinnamoyl-CoA reductase (CCR) pathway, which is involved in CoA ester to aldehyde conversion in monolignol biosynthesis that diverts phenylpropanoid-derived metabolites into lignin biosynthesis. RNA gel-blot analysis revealed that in wheat, *Ta-CCR1* is significantly expressed in the plant stem, with lesser expression in leaves, and unnoticeable expression in roots. By progressively increase in the CCR enzyme activity, lignin biosynthesis also went up along with stem maturity. Results suggested that up-regulation of *Ta-CCR1* mRNA levels and greater CCR enzyme activity were positively linked to high lignin contents in wheat stem with significant mechanical strength. This concluded that *Ta-CCR1* and its related CCR enzyme are intricate in lignin regulation during the process of stem maturity, thereby linked to stem strength support in wheat plant [96].

In rice, lignin biosynthesis pathway, its molecular and genetic regulation is not explicitly clear, just as in wheat. However, Ke et al. [112] conducted an experiment in rice to identify and characterize semi-dominant dwarf mutant (*pex1*) with stiff culm, and exposed that the phenotype of pex1 resulted from ectopic expression of *OsPEX1*: a leucine-rich repeat extension-like gene. An important point of this research was that a positive association between the *pex1* mutant and higher lignin contents was observed and it also enhanced the expression levels of lignin-related genes. Results summarized that the *OsPEX1* gene intercedes lignin biosynthesis and/or accumulation in rice, whereas transgenic rice cultivar without semi-dwarf alleles but expressing the *OsPEX1* gene showed a reduction in plant height, so we can conclude it has a key role to enhance lodging-tolerant and regulating lignin biosynthesis in rice. To comprehensively understand the molecular and biological cellular pathway, recent progress in CRISPR/Cas9-mediated genome editing could be a vital tool, as lignin-enriched transgenic rice by using CRISPR/Cas9-targeted mutagenesis of the transcriptional repressor *OsMYB108* was achieved. Rice mutants of *OsMYB108*-knockout revealed a considerable rise in the expression levels of lignin biosynthetic genes by enhancing lignin deposition in culm cell walls [113]. Hence, this study suggested that scientists should have to further investigate the regulation and complexity of biochemical pathways that counter lodging problem.

Moreover, some physiological processes such as high photosynthetic rate, photochemical efficiency of photosystem II, leaf water potential, stomatal conductance and activities of the antioxidant enzymes, and osmolytes consistency are closely correlated with root length and root length density to maintain water uptake by means of plant mechanical wounding to better perform under waterlogged conditions. Mechanical wounding also enhances lignin deposition and activity of those enzymes which play a role in synthesizing lignin contents to make plants more robust against lodging [114]. However, in cereals (such as in wheat and rice), more attentions from biologists and physiologists are still needed to investigate more profoundly on the subject; how plant physiological activities, and biochemical changes are linked to lodging stress and how could cell signaling molecules either regulate lignin biosynthesis up or down?

## 3. Plant Hormones in Lodging Stress as a Key Regulator

Plant hormones serve as a key element to regulate many plant traits which have a crucial role in lodging tolerance, for instance, tillers per plant. It is an important factor to control root lodging in wheat plant [115,116,117]. However, the researchers have no detailed knowledge and understanding about the underlying mechanism between hormones and wheat tillers, which trigger through respective exogenous hormones. Cai et al. [115] investigated the effects of indole-3-acetic acid (IAA) and zeatin (Z) applications on the wheat tillering and explored the subsequent mechanism which regulate the tillering occurrence in wheat. They found IAA repressed the rate of tillering, whereas external Z application stimulated the occurrence of tillers under the condition of low nitrogen. Moreover, the application of exogenous IAA entirely inhibited tillers bud growth, while Z hormone substantially accelerated the buds’ growth level in low nitrogen conditions. Their experiments concluded a considerably positive correlation between tiller buds’ growth and content of endogenous Z hormone. Whereas, insignificant weak association have been observed between tiller buds’ growth, and exogenous IAA, gibberellins (GAs), and ABA contents. So, their results suggested that zeatin (Z) hormone has an indispensable role in regulating the tiller occurrence and its buds’ growth by influencing the Z content. Therefore, it plays a vital role to assist plants in stronger soil anchorage, and this trait ultimately makes them tolerant to lodging stress environment.

As we have discussed earlier, semi-dwarf plants are more tolerant to the devastating effects of lodging stress under extreme environmental conditions, such as winds and floods. To date, 70 mutants related to dwarfism have been reported in rice (*Oryza sativa*), and many of them have been characterized as gibberellic acid (GA)-deficient or GA-insensitive mutants. The phytohormone GA has a pivotal role in many developmental processes of plants, including shoot and stem elongation, and plant height [118,119]. We know that these traits are also linked to lodging stress. *OsWOX3A* is a GA-responsive gene and categorized into the USCHEL-related homeobox (WOX) nuclear protein family. *OsWOX3A* has an important key role in the development of tillers, lateral roots, and root hairs by producing a rice dwarfism phenotype. On the other hand, it was reported to be involved in negative feedback regulation of the gibberellic acid biosynthetic pathway [120]. In this scenario, its function was completely protected by the application of exogenous GA3 that implies how gibberellic acid (GA) is inevitably rescued plant dwarf phenotype by maintaining lodging-related characters throughout the developmental process in rice.

Another important primary signal to regulate and to avoid lodging in plants is “ethylene”. Along with ethylene, alterations in ABA, gibberellin, and auxin concentration are also essentially required to improve faster growth under water conditions [121]. The question is: how ethylene is involved in escaping flooding effects in the case of wheat and rice? In rice, which has wider genetic diversity and to avoid flooding damage, two mechanisms are involved: first, increased ethylene production up regulates the expression of snorkel1/2 (*SK1/2*) through *OsEIL1b/OsEIL2* binding to the *SK1* and *SK2* promoters, where it up regulates the GA contents to stimulate internode elongation in deep water rice [122]. Secondly, by negatively regulating the GA response through SUB1A, an Ethylene-Responsive Element Binding Factor (ERF) which restricts Slender Rice-1 (SLR1) and SLR-like1 (SLRL1) degradation to impede the stem elongation at the seedling stage during severe flooding [123]. To enlighten more about its function, a large body of physiologist’s research concluded in their experiments that rapid internodal elongation in rice occurs due to the involvement of ethylene and gibberellin. A rice AP2/ Ethylene-Responsive Element Binding Factor (ERF) gene, *OsEATB* (linked to rice tillering) was cloned from rice and its ectopic expression revealed cross-talk between ethylene and gibberellin arbitrated by the *OsEATB* gene, probably trigger internodal elongation differences in rice [124]. As it is obvious that plant height is highly negatively associated with tiller numbers, we can thereby interpret ethylene’s role to obtain dwarfism and to escape lodging in important cereal crops like rice.

Moving to the next primary attribution of ethylene for aerenchyma formation in roots, which has been considered decisive for roots function under oxygen-deficient soils, this trait supports the plant for flooding tolerance. A positive association has been reported between aerenchyma formations in adventitious roots and shoots growth under hopoxia [74]. Further to this, aerenchyma is positively linked to wheat yield in the waterlogging situation. Some previous results also witnessed that ethylene signaling in rice not only triggers aerenchyma development in primary roots together with its involvement in the regulation process of constitutive aerenchyma, but also plays a role in root elongation [125]. To enlighten the crucial role of ethylene in the complex lodging mechanism, one example related to study in *Zea mays* plant where Shi et al. [126] investigated the effects of ethylene on the development of nodal roots, which are considered to be responsible for root-lodging resistance. Results revealed that ethylene precursor 1-aminocyclopropane-1-carboxylic acid (ACC) strongly support nodal roots emergence in maize. Further study of the transcriptomic analysis showed that genes expression which is involved in metabolic processes and cell wall biogenesis went up under the treatment application of ACC [126]. This implies the notion that ethylene is positively associated to regulate the outgrowth of young root primordial to ensure root-lodging resistance in maize.

Divte et al. [127] conducted an experiment in wheat, looking at how ethylene regulates root growth and phytosiderophore (PS) biosynthesis defines iron deficiency tolerance. We concluded from their research discussion that ethylene is an important factor in determining root architecture and traits like root growth, root hairs formation, elongation, and cluster root formation. These characteristics are also linked to biotic stress tolerance such as wheat lodging [127]. Despite the fundamental role of ethylene and its precursor in root lodging in cereal crops, for instance in rice and maize, it is also a prerequisite to conduct experiments in the wheat plant to explore the further role of ethylene in lodging.

## 4. Genetic Manipulation for Lodging Resistance

### 4.1. Quantitative Trait Loci

Several quantitative trait loci (QTLs) revealed resistance to lodging and its related traits have been reported in rice, wheat, and barley [30,59,128,129]. Identification of genes conferring resistance against lodging is required for enhancing cereal crops production in lodging-affected areas [19]. Conventional breeding techniques combined with recent advancements in biotechnology and genomics have the potential to identify and transfer lodging-resistance genes for enhancing the physical strength of wheat and rice crops and to increase the grain production. In addition, QTL mapping of lodging- resistance genes and their transfer to lodging susceptible genotypes provide the opportunity for cost-effective and to increase the grain production of wheat and rice in lodging prone zones.

The QTL mapping studies (Table 2) have been conducted on populations developed from Wheat x Spelt crosses, which exhibited extraordinarily high lodging resistance [130]. Keller et al. [131] reported lodging- resistance QTLs by using RFLP-based markers (Restriction fragment length polymorphism) in a Wheat x Spelt population. Three QTLs for culm thickness and five QTLs were identified that could reduce plant height. Composite interval mapping established the QTL for lodging resistance on chromosomes 4B (related to Rhi-Bl), while QTL for plant height was observed on chromosome 4D (related to Rht-DJ) [132]. Hai et al. [133] stated that two QTLs *QSs-3A* and *QSs-3B* were associated with plant stalk strength, whereas two QTLs *QPd-1A* and *QPd-2D* had been controlling pith diameter in wheat. They further investigated that one QTL (QSd-3B) was associated with culm diameter and one QTL (*QCwt-2D*) was controlling the expression of culm wall thickness in wheat. Additionally, SSR markers GWM340 and GWM247 were associated with a single strong stem QTL on chromosome 3BL in a wheat population derived from the cross Xinongshixin (strong stem) x Line 3159 (hollow stem) [37]. The QTL regions that have impacted the highest number of lodging-related traits of wheat tended to comprise the plant height (1D, 2A, 3A, 4A, 6A, 7D), culm diameter (3A, 4D), stem material strength (3A), and stem failure moment (1A, 6A), as presented in Table 2 [30]. These QTLs have sufficient meaningful effects on lodging risk in wheat and will be needed for further validation and fine mapping. *Rht5*, a dominant gibberellic acid-responsive dwarfing gene of winter wheat, has been identified to be located on chromosome 3BS and linked with the molecular marker Xbarc102 [134]. A major QTL (*QPH.caas6A*) of plant height was identified between Xwmc256 and Xbarc103, which had accounted for 8.0% to 10.4% of the phenotypic variation across eight environments using an F_2:4_ wheat populations derived from the cross between Jingdong 8 and Aikang 58 [135]. In a previous study, *QPH.caas-6A* was detected for plant height, which also associated with marker Xwmc256 on chromosome 6AL and accounted for 6.3–29.1% of the phenotypic variation [136]. To reduce plant height, more recently, *QPH.caas-6A* designated as *Rht24* (Table 3) was described as a novel Rht locus located on chromosome 6A and flanked by the two SSR markers, Xwmc256 and Xbarc103 (Table 2) [137]. This region showed highly phenotypic variations and could be validated to further fine-mapping studies in wheat. A QTL for plant height on chromosome 6A has also been reported by a few QTL mapping studies in wheat [136,138].

For determining lodging resistance in rice, two near isogenic lines (NILs) were selected from a series of chromosomal fragment substitution lines derived from the cross of two rice varieties, namely Kasalath and Nipponbare [51]. They reported that isogenic line NIL114 has a QTL of single culm diameter like *sdm8* while isogenic line NIL2 has four QTLs of culm diameter such as *sdm1*, *sdm7, sdm8, and sdm12*. Yano et al. [128] reported a new QTL for culm strength conferring resistance against lodging in indica population Chugoku117. Ookawa et al. [52] identified a new QTL *SCM2* regulating culm lodging resistance in a high grain-yielding indica cultivar Habataki. The *SCM2* was deployed into Japanese elite cultivar Koshihikari and near isogenic line NIL-*SCM2* was developed with culm lodging resistance and grain production. Guo et al. [139] isolated QTL *SCM3* and reported that *SCM3* encodes the product of a rice gene Teosinte Branched 1 (*OsTB1*)/*FINE CULM 1* (*FC1*) which was considered to have a key role in stri-golactone signaling. Near isogenic line having *SCM3* exhibited higher culm strength and produced a higher number of spikelet despite the decrease in tiller number [128]. Incorporation of *SCM3* and *SCM2* in a single population produced robust culms and enhanced grain production in comparison with Koshihikari cultivar. The QTLs viz., *prl5, SCM1, SCM2, SCM3*, and *SCM4* showed a positive association with culm resistance to lodging was transferred from the Moroberekan parent [52,128].

Five novel QTLs were reported in a rice population on chromosomes 4, 5, 6, 11, and 12 which conferred resistance in the basal portion of the plant to lodging, derived from the cross of japonica Nipponbare x indica Kasalath [140]. Six QTLs for culm diameter were identified on chromosomes 1, 3, 6, 7, 8, and 12. One QTL on chromosome 6 and one on chromosome 5 were mapped for lodging resistance to a heavy storm, respectively. Twelve QTLs regulating culm diameter, culm strength, and culm length were located on four various chromosomes 1, 2, 6, and 7 in the backcross population of Swarna 3 x Moroberekan [59]. They also reported the presence of three QTLs (*qCL1.1, qCL2.1,* and *qCL7.1*, are needed to explore the region by fine-mapping) for culm length on chromosomes 1, 2, and 7. One QTL (*qrl5*) for root lodging resistance has also been reported [105].

Furthermore, six additive QTLs for culm length were detected on rice chromosomes 2, 3, 4, 5, and 6 [141]. Four major QTLs for culm length were mapped on rice chromosome 1, 2, 5, and 6. One QTL (*qCL1.1*) for culm length was identified at the physical location of 36 M bp on chromosome 1 and accounted for 26% of phenotypic variation [142]. In addition, three QTLs associated with culm diameter were identified on chromosomes *qCD1.1* (1), *qCD2.1* (2), and *qCD7.1* (7) explaining 10–14% of phenotypic variation [59]. Six QTL, namely *CS1.1, qCS2.1, qCS2.2, qCS2.3, qCS6.1*, and *qCS6.2* on chromosomes 1, 2, and 4 were identified for controlling culm lodging resistance. The effect of deep-rooting Dro1-NIL on lodging resistance has been investigated in the study of Arai-Sanoh et al. [143]. However, the penetration ability of roots QTL Dro1 is still unclear. Thus, further analysis using the other genetic backgrounds will be needed to understand the relationship between deep rooting by DRO1 and its pleiotropic effect.

### 4.2. Gene’s Associated with Lodging Tolerance

The Indian Council of Agricultural Research successfully utilized the advantages of the green revolution through the introduction of cultivar Norin-10, Gaines pedigree cultivars, and locally adapted common wheat genotypes having *Rht* genes (Table 3) from Mexican varieties. This was resulted in to produce shorter wheat genotypes which imparted the lodging resistance, high grain yielding, enhanced uptake of nutrients and increased N fertilizers responsiveness attributes to these genotypes [144]. The most frequently used dwarfing genes, *Rht1 (RhtB1b)* and *Rht2 (Rht-D1b)* [145,146], were introduced from Norin 10 (Japanese variety) [147]. Shortening of plant height, in particular, with the effect of major dwarfing genes *Rht1* [148] and *Rht2* was observed in spring wheat [149]. These dwarfing genes reduced the number of internodes in relation to gibberellic acid: a plant hormone which stimulates growth and development [150,151]. Moreover, *Rht1* and *Rht2* genes decreased plant height by 14–17% independently from each other and shortened the plant height by 42% accumulatively in Germany and UK wheat varieties [152,153]. Another dwarfing gene *Rht3* reduced plant height by 59%, but has not yet been used in commercial wheat varieties [154].

*Rht5*, a gibberellic acid-responsive dwarfing wheat gene significantly decreased plant height by 25% to55% without decreasing seedling vigor and coleoptile length, as reported in References [155,156,157]. Other researchers reported that *Rht5* considerably reduced plant height but delayed the flowering time by about 4.8% to 14.0% in a thermal environment [28,155,156,157]. They described that *Ppd-D1* also reduced plant height at about 10%, while the combination of both *Rht5* and *Ppd-D1* generated even shorter plants by 45% and had achieved shorter genotypes with greater lodging resistance. Several genes, such as *Rht7* [158] and *Rh8* [159], *Rht-B1*, *Rht-B1a, Rht-B1a, Rht-D1, Rht8, Ppd-B1,* and *Ppd-D1* were associated in reducing plant height in wheat [160]. The semi-shorter phenotype of wheat lines (e.g., contain *Rht8* gene) was due to shorter inter-nodal length along the thicker culm, achieved through decreased cell elongation [161,162]. Additionally, gibberellic acid-sensitive dwarfing genes *Rht8* and *Rht9* have been transferred into several cultivars on account of superior early vigor in dry environments. These dwarfing genes (*Rht8, Rht-B1b,* or *Rht-D1b*) had a slight shortening influence on the coleoptile length, which led it to either spread or grow deeply at high soil moisture levels and ultimately reduce plant height [163,164]. These genes have been commonly used to decrease height of wheat cultivars in Eastern and Southern Europe, China, Japan, and Russia [165]. Gibberellic acid-responsive height shortening genes encompassing *Rht4, Rht5, Rht8, Rht9, Rht12,* and *Rht13* have been reported in bread wheat [134]. Several plants dwarfing genes containing *Rht14, Rht15, Rht16, Rht18, Rht19,* and *Rht-R107* were also reported in durum wheat [166,167]. Recently, Vikhe et al. [168] reported that the *Rht18* gene was associated in reducing plant height in durum wheat. Genes associated with plant height reduction containing *Rht22, Rht-B1IC12196,* and *Rht-B1f* were identified in *T*. *turgidum*, *T*. *polonicum*, and *T*. *aethiopicum*, respectively [166,167,169]. The dwarfing genes *Rht14, Rht16,* and *Rht18* were associated with the reduction of plant height in durum wheat [167]. Ma et al. [170] reported that the COMT gene *W-cm5-1* was expressed in root, plant stalk, and leaf tissues in wheat and the transcriptome level of the *W-cm5-1* gene in the up-growing wheat stem was accompanied with the stem thickness and tolerance to lodging stress.

Several improved semi-dwarf varieties of rice have been introduced and grown throughout the world. However, despite their short-statured nature conferred by the semi-dwarf1 (*sd1*) gene [171,172], lodging happens in several rice cultivars during robust typhoons smash in Eastern and Southeast Asian countries. Almost 53 semi-dwarf maximum yielding rice cultivars were collected from the USA, China, and Japan, among them 38 cultivars were found to contain an sd-1 allele (Table 4) [171]. The *sd-1* gene has been utilized in several breeding programs to decrease plant height and has substantially improved lodging tolerance in rice [41]. Ashikari et al. [173] reported that dwarfing gene *sd-1* was deployed into improved rice lines to produce semi-dwarf rice variety. Several genes are involved in the biosynthesis of the primary and secondary cell wall. Some genes like *OsCesA1* (Cellulose synthases), *OsCesA3,* and *OsCesA8* genes are expressed in the primary cell wall [174] while other genes like *OsCesA4, OsCesA7,* and *OsCesA9* are predominantly expressed in secondary cell walls [38,175]. Some polycomb group genes mutants showed a shorter phenotype in rice [176]. These polycomb genes containing *OsCLF, OsEMF2b,* and *OsFIE2* reduced plant height when they down regulated their expression. Furthermore, the *OsEMF2b* mutant *emf2b* showed a reduced plant height [177,178]. Similarly, *OsFIE2* RNA interference (RNAi) transgenic genotypes and mutant fie2 also exhibited a dwarf phenotype [179].

Furthermore, lignin enhances the stem mechanical strength by involving in cell wall biosynthesis. The expression of *PAL, CCoAOMT, CCR, COMT,* and *CAD,* which are involved in lignin biosynthesis are accompanied with culm strength and resistance to lodging [180]. The role of *COMT1, CCR1,* and *CAD1* genes in enhancing stem strength against lodging is greatly accompanied by stem lignin accumulation and the strength of mechanical tissue [37,96]. Similarly, the *TaCM* gene involved in the biosynthesis of lignin content was related to the stem rigidity, root tissue, and lodging tolerance [95].

## 5. Agronomical Management

### 5.1. Sowing Time

Sowing time is one of the most important factors affecting lodging resistance in cereal crops. So, selecting an appropriate sowing time is considered to be a vital perspective to reduce lodging threat in winter wheat [181]. It has been witnessed that late sowing substantially reduced the threat of lodging in wheat, particularly by shortening the internode lengths, plant height, and culm length at the center of gravity, and via increasing culm wall thickness, diameter, and grain filling duration [182]. For example, delayed sowing by only two weeks could decrease the risk of wheat lodging by up to 30% [183]. Further to this, earlier winter wheat sowing resulted in a higher number of extended internodes and could increase the incidence of stem base infection such as Fusarium foot rot [184], which, therefore, enhances the chances of lodging occurring by a weakened stem base [30,185]. About 71% of plant height differences among wheat plants were manifested using various sowing times on account of the number of extended internodes [185]. Moreover, Kirby et al. [186] reported that wheat cultivated during the first weeks of September and December showed heights of 94 cm and 66 cm with 6.2 and 4.8 extended internodes, respectively. Substantial increases in grain yield could be possible from even earlier planting but this would require appropriate cultivars with maximum resistance to diseases and lodging [187,188]. Cultivating rice after the optimum sowing time increased the insect and disease incidences, tropical storm-related lodging, and cold stress damage during heading and the grain filling with reduced grain production [184,189]. On the other hand, late cultivated cereal crops are generally shorter than the ones cultivated earlier [2,182,190]. Reduced plant height is often accompanied by a decrease in the number of extended internodes [191].

### 5.2. Sowing Depth

Deep sowing enhances the depth at which the root crown is located along with its length [192]. This increased sowing depth makes the stronger anchorage of the plants into the soil, thereby enhance the lodging tolerance. Even deeper sowing has been found to play a role in reducing lodging [25], however, until now published articles’ data have not reported sowing depth effects on lodging widely. Moreover, it was reported that the direction of drilling had no effect on the severity of lodging [193]. Deeper drilling helps in adjusting the depth of crown roots of plants to a depth of 40 cm. Hence, it is better to sow within a 4–7 cm distance. Shallow drilling more than 4 cm may be expected to raise the crown and its structural roots, as a result, weakening the mechanism of anchorage [194]. Along with the sowing time, sowing depth also affects the resistance capacities of wheat and rice plants to lodging [191]. Sowing depth considerably impacted the emergence and vigor of seedlings contributes greatly to crop stand and grain yield [191,195,196]. Ali et al. [197] had sown three wheat varieties as broadcast at the medium rate from 2003 to 2006. Field experiments comprising three wheat varieties sown at different depths revealed that the relatively deeply sown seeds up to 6–7 cm had better soil anchorage with reduced lodging [194,196]. Similarly, plant height was affected by deep planting and reduced after deeper sowing of wheat seeds up to the optimum level, which indicated less lodging [195]. In conclusion, deep sowing in wheat at optimum levels resulted in reduced growth and tillering, which affected the plant canopy with better plant anchorage and minimized the possibility of lodging. In contrast, transplantation of rice seedling at deeper depths could result in delayed and retarded plant growth and consequently lead to poor crop stand [198]. In case of direct seeding, rice seeding should be no deeper than 2.5–3.0 cm so that rice coleoptile could emerge properly [198].

In addition, among the three cultivation methods (mechanical transplantation, dry direct-sowing and wet direct-sowing), rice sowed with mechanical transplantation method under shallow depth showed the strong resistance to lodging [172]. However, they also reported that the average lodging index of the two basal internodes was reduced by 16.08% and 24.69% under mechanical transplantation in comparisons to dry direct-sowing and wet direct-sowing. Similarly, the first, second, and third basal internodes of rice plants under the mechanical transplantation substantially increased the breaking resistance of the stem and bending moment of the rice panicle, but greatly decreased the lodging index, which benefited from their thicker culm wall, larger culm diameter, greater biomass accumulation, and greater lignin accumulation in the stem [199,200]. Moreover, shallower sowing in comparison to the aforementioned level could produce seed drying prior to the emergence. Direct-seeded rice is more prone to lodging in comparison to transplanted on account of its higher plant population. Furthermore, in direct-seeded rice, the base of the plant is above-soil with poor anchorage [201,202]. Ahmad and Mahmood [203] reported that less impact of lodging was observed in bed planting (20.5%) in comparison to wheat cultivated by flat method (34.6%). Minimum lodging in raised beds was due to drainage of excessive rain water from the fields and stronger plant anchorage on the beds, as a result, higher grain yield.

### 5.3. Planting Density

High plant populations have been widely used to enhance grain production in wheat and rice. This, however, results in a high risk of lodging and crop yield losses [199]. Increased wheat population (planting density) results in a greater competition between plants for light and nutrients, which may weaken the diameter and number of nodal roots [204], as a result, reduces the lodging resistance [205]. In addition, high seed-density will also make a greater chance of lodging stress by accelerating culm length and lessening culm diameter along with total root mass [193]. Due to dense population, plant anchorage strength was reduced because it caused the lesser spread of the root plate [77]. In contrast, lower plant density and less dense canopy reduce disease build-up and vulnerability to lodging in wheat [187,206]. The intensity of lodging was enhanced in wheat due to increasing plants from 150–400 plants/m^2^ [11]. These factors reduced the lodging risk due to minimizing plant density from 400 plants/m^2^ to 100 plants/m^2^. Hence, maintaining a lower plant population results in more crown roots and thus, better anchorage [194].

Guyer and Quadranti [207] reported that in most of the cases, seed rate is applied up to 200 kg ha^−1^ to plant a dry-seeded rice crop. Usually, the use of higher seed rates per unit area is applied to achieve getting a higher number of panicle and increased grain yield [208]. However, maintaining dense plant population at high seed rates could lead to promising environments for disease (e.g., sheath blight) and insect attacks in rice (e.g., brown plant hoppers), making plants highly vulnerable to lodging [149]. The biggest disadvantage of the densely planted rice field is the highly interspecific (with wild relatives and weeds) and intraspecific competition among the plants, which results in continuous shading and lodging resulting in straw production instead of grains [209,210]. The influence of spacing on the grain production and lodging resistance was studied using tall indica rice cultivars and the spacing of 20 cm × 10 cm highly contributed to the grain production in the medium- and late-planted rice cultivars [211]. In addition, reduction in the plant density of rice is beneficial to increase culm diameter and reduce the length of basal internodes, dry matter weight per unit internode, and culm wall thickness resulting in increased breaking resistance and reduced lodging index [139,210].

### 5.4. Irrigation Method

Lodging causes the stable displacement/movement of plant stalks from their vertical position under storms. The huge amount of irrigation or rainfall increases the lodging strength of the soil. This prominent moist condition makes the soil more vulnerable to reduce the anchorage strength of the plant, and ultimately lead to plant lodging stress [11]. The risk of root lodging increases when the soil surface is heavily moisture saturated [73,193]. Root lodging happened more frequently when the plant had a heavy spike or panicle in waterlogged soil after irrigation [9] or rainfall [73]. Moreover, a portable wind tunnel was used in the field and root lodging happened only when soil contained high moisture content [212]. Lodging was forced due to high-velocity winds in February–April coupled with rainfall particularly in February and March at the milky stage of the wheat crop [213]. The situation was further enforced due to silty and clay soil, which made a temporarily waterlogged condition and this, favored the root lodging of the crop in wheat [214]. In wheat, lodging occurred on account of being highly moisture-saturated in the irrigated environment and it was concluded that there was always a root lodging threat in the fields when the upper land surface and lower soil were moisture-saturated [13]. Peake et al. [215] reported grain production loss of 1.7 t ha−1 due to lodging in the irrigated spring wheat of subtropical Australia.

In cereal crops, sprinkling irrigation could promote lodging at the early vegetative developmental stages. Water supply through sprinkler irrigation favored lodging when practiced at preliminary developmental stages in thickly planted wheat genotypes [216]. They also recommended that supplying water by sprinkler irrigation in space-planted genotypes should be withheld until plants reached the booting period. Ma et al. [217] stated that by an appropriate degree of regulated deficit irrigation from the period of the beginning of spring growth to the end of the stem elongation stage of wheat shortened the length of the first and second internodes, and increased their weight per unit length, and thereby enhanced stem lodging resistance. Moreover, flood irrigation could make the soil surface softer with a negative impact on the anchorage of roots in soil [15].

Root lodging is caused by heavy wind action into the canopy (wet soil) – forces which can cause root lodging in which the root system rotates in the soil [34]. However, heavy and continued rainfall may also cause failure to the root system, because of deep moist soil increasing the soil weakness to a point where structural roots lose anchorage in the soil, and even a light wind may exert a sufficient torque to induce root lodging in wheat [25,218]. In heavy irrigated soil the roots and crowns were rotated due to heavy wind, and as a result, root lodging was developed in cereal crops [194]. In contrast, dryness and/or cracks of the upper soil may restrict the development of the coronal root system and thus, promote lodging [194]. However, Griffin [219] reported that root lodging was often associated with the development of cracks in the soil on the opposite side of the plant to the lodging in wheat.

### 5.5. Effect of Nitrogen

Regarding nitrogen (N) concentration, the period of the N fertilizer application also affects the lodging score of wheat and rice [46,48]. Culm strength or thickness plays a critical role in imparting resistance to lodging as it gives a better control to hold the plant upright [26] and because culm breakage usually occurs at lower internodes [36]. Kong et al. [220] observed that mostly culm breaking happens at lower or basal internodes due to high NH+ application in the wheat plant. Similarly, a substantial enhance in lodging score was detected in wheat when N fertilizer was broadcast before the culm elongation phase [46] while no noteworthy influence was reported beyond the flowering stage [221]. Crook and Ennos [222] reported that at high levels of N application (approximately 240 kg ha^−1^) in wheat, the root and stem were 17% and 20% more prone to lodging respectively, in comparison with the application of 160 kg N ha^−1^. Similarly, Kong et al. [220] reported that the application of high NH^+4^ substantially reduced culm thickness. The low N application resulted in increased water-soluble carbohydrate concentrations (25% in the peduncle, 21% in the middle internode, and 42% in the lower internodes) compared to high N in wheat [89]. Morphological traits associated with lodging of rice genotypes increased with the increasing of N rate under low soil fertility [223]. They suggested that high N application results in enhanced vegetative growth that reduces the root penetration in the underground soil, which in turn inhibits plant root anchorage in soil, thereby, increasing the lodging risk. High amounts of inorganic fertilizers containing N decreased the thickness of the stem cell wall components (especially lignin content) in the early sown wheat [57,224,225]. Zhang et al. [44] reported that constituents of plant cell wall thickness were decreased by greater N application in rice. Furthermore, high N levels decreased lignin concentration in sclerenchyma tissue accompanied by the weakening of sclerenchyma cell wall strength [44]. A recent report by Shah et al. [191] stated that even though N-deficient genotypes had thin basal internodes and thin stem wall, these genotypes were observed to be least prone to lodging in comparison with the genotypes provided with sufficient N. Several studies had shown that wheat [56] and rice [226] genotypes provided with higher N levels had high lodging index on account of reduced lignin accumulation, thinner stems, and enhanced lodging angle growth pattern. Strong stemmed genotypes with enhanced stem thickness could be developed with low levels of N application in wheat and rice [222,227]. Berry et al. [77] also reported that lowering N application in spring wheat could decrease the height of the plant but would improve stem strength with enhanced stem diameter and wall width. Plant cell wall components including lignin and cellulose contents enhanced at first and then reduced progressively with the increasing N amounts in wheat [224]. Numerous studies have shown that lodging reduction has been attained at the cost of sacrificing yield potential by reducing N concentration and delaying N fertilizer applications [77,228].

### 5.6. Effect of Potassium

Nitrogen (N), phosphorus (P), and potassium (K) are macro-elements essential for plant growth and development [229]. The association between N and K has a pivotal role towards the improvement of crop grain yield and quality in rice [229,230]. However, elevated K^+^ levels along with high NH_+4_ could account for reducing culm diameter. For example, a grain yield reduction of 30–35% in rice due to lodging could be associated with the provision of higher levels of N and P fertilizers in the absence of K [231]. Imbalanced applications of N and K produced a taller rice genotype, which was highly prone to lodging as compared to semi-dwarf varieties. In rice plants, stem lodging could occur at three sites [231]: from the base [232], at the inter-nodal portion, and heavy panicles [172]. Moreover, the breaking strength of the stem was enhanced by K application response in rice [229] and wheat [233]. Balanced application of N and K positively enhanced the root growth and better root anchorage with soil-reducing lodging incidence in rice [231].

High amounts of K^+^ culm constituents were strongly associated with culm strength and lodging resistance in rice as optimum K^+^ nourishment was correlated with the lignin deposition into vascular bundles and sclerenchyma cells of the cell wall [37]. Similarly, K^+^ substantially decreased the negative effects associated with higher concentrations of NH_+4_, resulting in about 20–27% enhancement in culm mechanical strength at the time of grain filling and about 34.6% increment in the N remobilization efficiency in wheat [220]. Furthermore, it is extensively recognized that K^+^ has a major role in the photosynthetic process and metabolism of the resultant carbohydrates in plants [234,235]. Kong et al. [220] reported that K is tangled in the cellulose and lignin contents of the cell wall providing stem strength in wheat.

### 5.7. Effect of Silicon

Silicon is also a pivotal factor for rice growth and development. It has a positive impact on plant height, lodging resistance, inter-node length, bending strength, and enhances tolerance capacity to lodging in rice [236,237]. Application of silicon-enhanced lodging-associated traits like panicle length, stem length, plant height, and the third internode length, but the fourth internode length was decreased which is vital for resistance against lodging [238]. Silicon solution significantly reduced lodging percentage of rice by enhancing stem breaking resistance and inter-nodal length [239]. It was revealed that the silicon application could be distributed between the third and fourth internode, which enhanced the lodging resistance [238].

In previous rice and wheat studies, resistances to lodging in response to the dressing of soil with low and high amounts of silicon have also been reported [240,241]. Accumulation of silicon element in rice shoots increased the cell wall thickness of stem and the size of the vascular bundles [242] with the reduced lodging index. Strong storms causing lodging also enhance evaporation of water from leaves and dehydrate the plant tissues [243]. The application of silicon prevents these water losses [244]. Moreover, this element also affects the architecture of plant leaves to increase the amount of light interception by leaves [245], thereby improving sclerenchyma, vascular tissues, and vascular sheaths [246,247], and preventing lodging in wheat and rice. This micronutrient can be utilized to stimulate silicification and lignification in thick-walled cells, thicken collenchyma cells, and improve keratinocyte development and enhance cellulose content, as a result increasing lodging resistance in rice [248].

### 5.8. Lodging-Resistant Cultivars

Agronomists categorized the rice and wheat genotypes into three groups based on the response of the genotypes to lodging stress: highly, moderately, and susceptible [50]. Several wheat genotypes (Table 4) e.g., UP 2338; Munia/Kauz PBW 343, Seri 82, Star, Baviacora 92, and Weaver exhibited a crop falling score of less than 10 and were considered as lodging-resistant genotypes [249]. Several studies have depicted that Baviacora-92 is a wheat variety with a single dwarfing gene and is comparatively more lodging-resistant than those varieties which have two dwarfing genes [250]. Some of the wheat varieties e.g., Rialto, Buster, Hereward, and Savannah were more resistance to root lodging than the stem lodging [47]. While wheat varieties which were superior in resistance to stem lodging in comparison to root lodging includes: Spark, Cadenza, Mercia, Hereward [222], H4564 and C6001 [170], and Yangmai 20 [44].

Additionally, resistance to root lodging in four wheat genotypes: Riband, Hereward, Widgeon, and Galahad was due to the lesser “self-weight” moment of the stalk with impact on the capacity of the root system to resist the overturning moments [34]. Crook and Ennos [222] reported that genotypes resistant to lodging had robust anchorage that could resist the self-weight moments produced by the stems. In contrast, they also reported that cultivars vulnerability to lodging either had weak coronal root systems either generating greater self-weight moments or poor anchorage resulting in the longer stems. Aikang 58, Zhoumai 22, Zhoumai 18, and Pingan are the wheat genotypes planted in China and they enhanced the resistance against lodging [251].

Kalyansona and Sonalika were semi-dwarf, lodging-resistant wheat cultivars developed in the late sixties and occupied the wheat growing zones particularly in the Indo Gangetic Plains of India [252]. Wheat cultivars such as Kohika, Oslo, Sapphire, AC Foremost, ND695, and AC Vista were semi-dwarf lodging-tolerant and introduced into elite western Canadian bread wheat germplasm to increase lodging resistance [27]. Similarly, the semi-dwarf wheat variety Norin 10, which has an important source of dwarfing genes to reduce plant height and still has been utilized in wheat breeding programs [253]. This variety was originated from the cross of Daruma (native Japanese dwarf wheat cultivar) with two American wheat cultivars, Turkey Red and Fultz [254]. The japonica rice cultivar Wuyunjing 23 is lodging-resistant on account of strong mechanical strength of the stem [38,46]. The indica rice variety Takanari has robust culm characteristics due to its large section modulus, which exhibits the strong culm thickness [129]. Rice grain yield across Asia was increased from 1–2 tons ha^−1^ through cultivation of the lodging rice-resistant cultivar IR-8, in comparison with traditional rice cultivars grown on irrigated soils [255].

However, rice cultivars such as IR-8 and other IR series were used as parents in rice breeding programs under the National Agricultural Research System, resulting in the development of high-yielding rice varieties with desirable attributes such as resistance to lodging and diseases [252]. Pusa Basmati, a high yielding dwarf aromatic basmati variety was developed using marker-aided selection. Similarly, basmati rice hybrids, Pusa-1121 and RH-10, were developed through the utilization of marker-assisted selection method [256]. Semi-dwarf japonica cultivars Dontokoi and IR24 having high grain yield and quality were released in Japan. Both the cultivars were lodging tolerant with high tillering capability under waterlogged conditions [257]. Similarly, the advanced line (T5105) of rice has a semi-dwarf phenotype with improved resistance to lodging and a superior harvest index, in contrast KDML105, cultivated in the same field, encountered greater lodging occurrence at the time of seed setting stage as a result of its tall plant type [258]. Two other lodging-resistant varieties Wuyunjing23 and Yliangyou2 produced about 12% and 31% higher grain yields when compared with lodging prone cultivars W3668 and IIyou084 at low temperature, respectively [15].

## 6. Chemical Management

Spraying plant growth regulators at the appropriate crop growth stage can improve stem strength, shorten plant height, and prevent lodging [11,259]. Plant growth regulators that prevent gibberellins biosynthesis are most commonly being utilized in high input cereal management to reduce straw, which also enhances resistance to lodging [259]. Previously, several plant growth regulators were substantially applied in several crops to reduce lodging through decreasing plant height and to achieve a stable development in grain yield [11,260]. The plant growth regulators (most commonly gibberellins inhibitors) which have been utilized to shorten the stem growth are the onium-type compounds which contain Mepiquat-Cl and Chlormequate chloride (2-chloroethyl-*N*,*N*,*N*-trimethyl-ammonium chloride, CCC). Mepiquat-Cl interferes with ent-kaurene synthesis while Chlormequate chloride acts at the primary stages of gibberellin biosynthesis (Rademacher, 2000) (Figure 4). Inhibition of the transmitting of Geranyl-geranyl diphosphate synthase (GGPP) into Copalyl diphosphate synthase (CPP) on the binding of CCC to the enzyme CPP-synthase decreases the availability of bioactive gibberellins [261,262,263,264]. Some plant growth regulators like N comprising heterocycles, e.g., imidazoles and triazoles [265] are also used to control lodging. These growth regulators affect the activities of paclobutrazol (PBZ) and are nearly associated with uniconazole-P [263]. The aforementioned compounds affect the oxidation of ent-kaurene to ent-kaurenoic acid [263], whereas the prevention of oxidation of mono-oxygenases occurs when the oxygen is displaced from the binding site of the enzyme at the proto-heme iron site making oxygenase non-functional. Several gibberellins inhibitors have cimectacarps (trinexapac-ethyl), react or interfere with the gibberellins metabolic activity by interfering with 3ß-hydroxylation of GA20 to make the bioactive GA1 [260,266,267].

The application of Chlormequat at 3–5 leaf stages in wheat decreased the plant height by 3.3–14.5% [268]. Similarly, Chlormequat decreased plant height by 1–29% [269], 6% [270], and 3–15% [7] (Table 5). When applied at second node detectable/third node visible and pseudo stem erection-first node visible, pseudo stem erection/first node visible respectively, in winter wheat. Treated with Ethephon and CCC, cellulose, hemicellulose, and lignin concentrations in the cell wall of the middle internode and lower internodes changed less than 5% in the wheat [89]. At the vegetative development of spring wheat, Chlormequat decreased plant height by 20 cm [271]. Leśniowska-Nowak et al. [272] stated that chemical Chlormequat inhibits the biosynthesis of gibberellic acid in plants and enhances the culm length at the period of shoot extension. The applications of Chlormequat and Ethephon in combination at stem elongation stage substantially decreased the plant height in wheat [273]. Ethephon decreased plant height from 2–12% (12 cm) along with enhanced grain production up to 5.4% in wheat [271,274]. The application of plant growth regulators like ethephon, Chlormequat chloride, trinexapac-ethyl, and their integrations had no effect on the head diameter of sunflower [275]. Head diameter of the plant was affected by the growth or developmental stages and the application of aforementioned growth regulators at the early growth stages (for example, two visibly extended internodes) were observed to be more efficient in comparison to those applied at the late plant growth stage [275].

Application of Paclobutrazol drastically enhanced lignin concentration in the stem cell wall and it is closely associated with enzymes functioned in the basal second internode [276]. It increased culm diameter, filling degree of the internode, and wall thickness with cumulative affect by escalating resistance to lodging in wheat [5]. The applications of Palisade and Manipulator separately, and in integration, reduced the plant height of the wheat and lodging index, and consequently enhanced the grain production. Moreover, these growth regulators increased the wall thickness and stem breaking strength [277]. The application of paclobutrazol (for example, PP333) improved the phenylalanine ammonia-lyase (PAL), tyrosine ammonia-lyase (TAL), cinnamyl alcohol dehydrogenase (CAD) activities, enhanced lignin content, breaking resistance and culm lodging resistance index, and thus reduced lodging in wheat [86]. Trinexapac-ethyl, plant growth regulator, has proven the efficiency to reduce plant height in wheat [272,278]. The application of trinexapac-ethyl to the wheat cultivars OR-1, CEP-24, and CD-104 at two different early growth stages (between the first and the second nodes and between the second and the third visible nodes) has reduced plant height in comparison to their applications at the later growth stages [279,280]. The mode of action of this plant growth inhibitor involves the modification and lowering the stability of gibberellins through inhibition of the 3 â-hydroxylase enzyme which in turn inhibits cell elongation [279].

The application of PGRs can increase wheat grain production [273], however, in some instances; the use of PGRs does not substantially affect wheat grain yields [280]. The application of paclobutrazol and Chlormequat can reduce the stem length and thus, increase grain production, while gibberellin has the contrasting effect [78]. The use of trinexapac-ethyl at a greater concentration decreased grain yields [278], while trinexapac-ethyl and Chlormequat did not affect the grain yield [281]. However, the use of Chlormequat and trinexapac-ethyl at mid tillering and early stem elongation increased grain yield by 4% and 8% respectively, compared with the control in wheat [282]. As gibberellins plays a critical role in cell division during stem elongation and vegetative growth stages, reduction in gibberellin levels resulted in retarded plant growth in rice [283]. The application of gibberellic acid inhibitors, especially paclobutrazol, affected different physiological and yield-associated traits in rice [284]. For example, the application of paclobutrazol to Japanese paddy rice reduced the plant height by 15–25%, enhanced the resistance to lodging by 60%, and increased the grain production by 15% [285]. Trinexapac-ethyl and Prohexadione-calcium have drastically decreased gibberellic acid biosynthesis resulting in a reduced stem length and decreased lodging index in rice [286]. Kim et al. [287] reported that the application of Prohexadione-calcium at early growth stages in rice was highly effective to reduce lodging in comparison to its application at later growth stages. Culm length and the third internode length were decreased while stem strength was enhanced by the application of Trinexapac-ethyl and Prohexadione-calcium.

Application of PBZ with 50 mg L-1 at the booting stage could enhance the number of spikelets per panicle, seed setting rate, and grain production (8–13%) in Huayou86 and Peizataifeng in both seasons in rice [288]. Similarly, application of PBZ on the leaves surface at the late growth period of the late-season rice could enhance seed setting rate and grain production by delaying leaves senescence [44]. Grain yield was increased 10% and 6% by spraying exogenous PBZ and GA3 at five days before the flowering stage, respectively [289]. Greater grain production was obtained when cycocel was sprayed, followed by nitrogen and K2O fertilizers [290]. The spraying trinexapac-ethyl significantly enhanced grain yield in rice [20,291].

## 7. Future Prospects

In wheat and rice, lodging drastically reduces both grain yield and quality. These crops are highly prone to lodging at late vegetative and reproductive stages. The impact of lodging regarding grain yield reduction varies from genotype to genotype, while a significant reduction in yield on account of lodging is usually observed under the condition of high input and mechanized agriculture. This review focused on emphasizing the presence of genetic variation among the wheat and rice genotypes for various plant traits controlling the lodging resistance. The lodging-induced damages in wheat and rice could be significantly reduced with more focus on these traits in crop breeding programs. But we realized, in terms of plant physiology, no significant research has been done precisely to understand the physiology parameters like photosynthesis, transpiration efficiency, and stomatal conductance that could be related to understand the complex lodging phenomenon not only in cereals plants but also in other crop plants, so it could be a vital topic of interest for breeders and physiologist to investigate further. From the morphological point of view, plants have an association between morphological traits and lodging-resistance and these characteristics are critical for conferring lodging resistance to crops. For example, stem lodging research has concentrated on stem diameter strength with the findings that a stronger stem is more resistant to lodging in comparison to a weaker stalk, regardless of others morphological characteristics, chemical composition, or weather conditions. However, sometimes it is argued that the estimation of these factors is not considered reliable for lodging resistance and hence, further deep attention is required to figure out other precise agronomical and morphological parameters that evidently confer lodging resistance.

As we know, lignin contents are the major constituents of the plant cell wall and recognized as indispensable components essential for plant vigor and play a role against biotic and abiotic stresses including plant lodging. Unfortunately, detailed dissection of the molecular mechanism of lignin formation has not been studied thoroughly. Therefore, it is suggested to researchers to design projects and conduct experiments to explore biochemical pathways particularly associated with lignin biosynthesis in wheat and rice. Moreover, they must design new tools and approaches to alter the quantity and quality of lignin in these staple food plants without disturbing its functions in conferring structural support for normal growth and development.

In terms of natural disasters like high frequencies of winds and rainfall can be a leading source of catastrophic crop lodging. Therefore, to escape the occurrence and devastating effects of lodging, deliberate efforts should be planned aiming towards enhancing the anchorage strength rather than stem strength. This could be approached in two ways: (i) breeding for rice plants along with stronger and stiffer roots, with no reduction in the existing number of roots per plant, and (ii) the soil medium mixture could be changed in such a way that would make it stronger to favor plant anchorage.

Proper agronomical managements (Figure 5) of wheat and rice crops related to use of lodging-tolerant cultivars, sowing time, planting density, irrigation methods, fertilizer applications, crop protection, and more importantly, application of plant growth regulators to manipulate plant height can reduce losses on account of lodging. Plant growth regulators can reduce the lodging susceptibility through modification of plant architecture, improvement of canopy structure, reduction of stem length, lowering the leverage of the ear and other upper plant parts. Future studies are needed to focus on investigating the impact of plant growth regulators on stalk and root anchorage. Moreover, it is inevitably indeed to develop and investigate new growth regulators that could strengthen lodging-resistant traits. The use of optimum levels of N, P, K, and silicon fertilizers are important in cereal crop production systems. Generally, P application enhances grain yield while K is important for flowering. The amount of green leaf area and efficient use of solar *radiation* input are important aspects in determining carbohydrate productivity. Appropriate plant canopy area is an essential feature of modern agronomy systems and is usually achieved through the use of balanced nitrogen.

Despite our current understanding of growth regulators and plant hormones associated with lodging, still many questions remain to be answered, for example, the role and the mechanisms of ethylene perception, signal transduction, and transcriptional regulation. Further characterization of ethylene’s character would assist agronomists and breeders to provide new insights to explore the complex genetic traits of lodging.

Moreover, there is a need for stringent research actions to group modern wheat and rice genotypes for lodging-resistant traits which could be used in the future as a basis for identification of QTLs conferring lodging resistance. It would be interesting to study and establish a linkage relationship between lodging-resistant QTLs and QTLs for other agronomic traits e.g., stronger stem and shorter plant stature could be linked to lodging-resistant loci. Thorough investigations are required to be performed on the development of reliable morphological and molecular markers for plant height, stem diameter, the spread of the root plate, the material strength of the stem wall, and stem failure moment to improve lodging resistance in wheat and rice. In conclusion, we also propose a novel high-throughput phenotyping (HTP) approach to understand the underlying complex genetic architecture of lodging and to understand genotype-to-phenotype relationship to accelerate the plant breeding program. It would enable us to accurately measure huge populations under field conditions that are required for genomics studies and breeding progress.

## Figures and Tables

**Figure 1 ijms-20-04211-f001:**
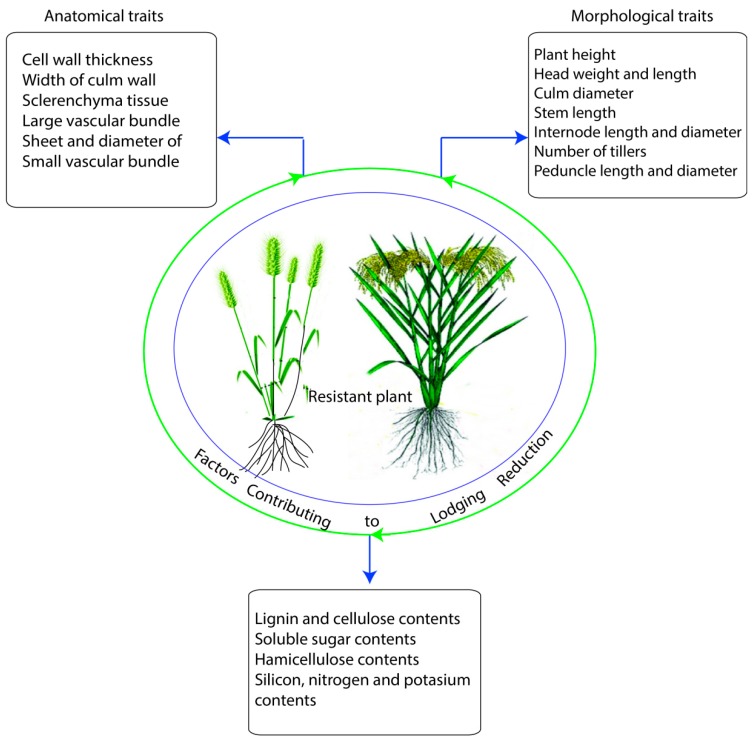
Major traits associated with the reduction of lodging in wheat and rice. Factors associated with the lodging resistance are shown by (blue) arrows, whilst resistant plants through these contributing elements are encircled in (green) arrows.

**Figure 2 ijms-20-04211-f002:**
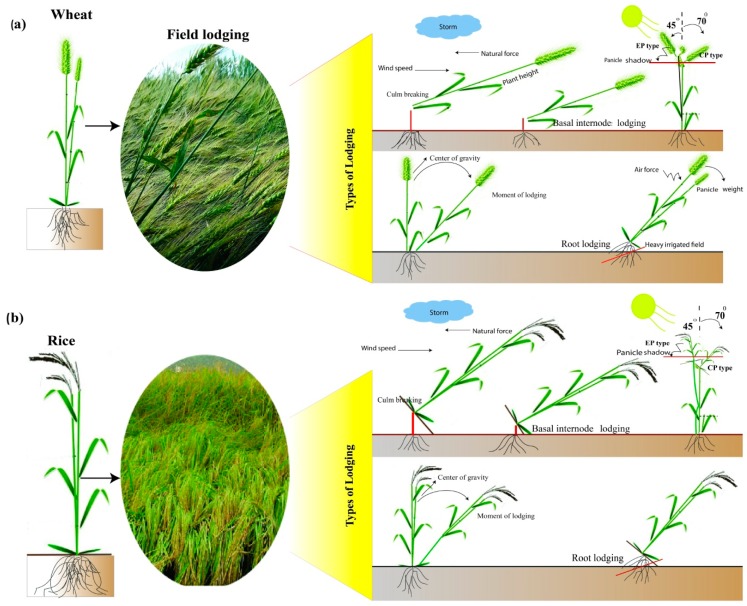
Various types of lodging that are devastatingly reducing yield in (**a**) wheat, (**b**) rice: diagrammatically presented as follow: culm breaking; basal internode lodging; panicle shadow on a leaf during sunlight and the force of the panicle on the stem between erect (EP) and curve panicle (CP) plants; lowers the center of gravity due to decrease in plant height and increases bending type lodging resistance, and root lodging.

**Figure 3 ijms-20-04211-f003:**
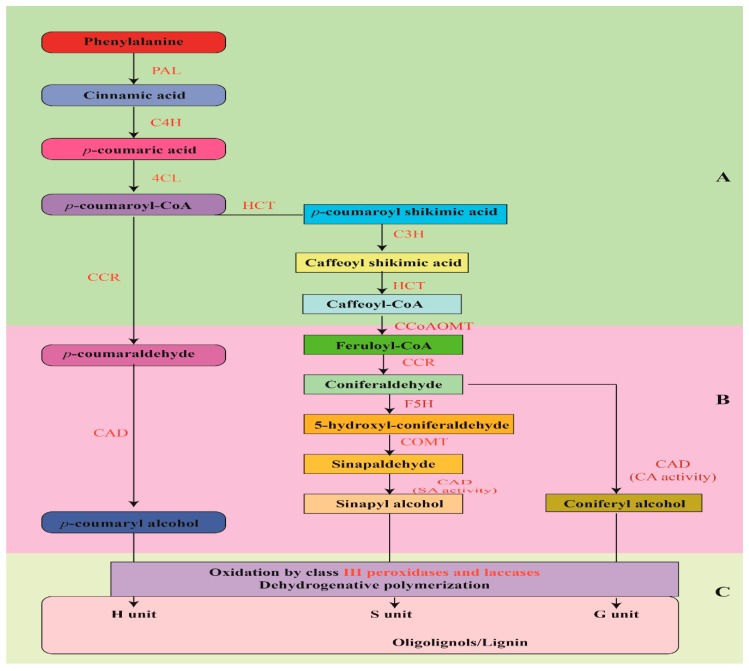
General steps involved in lignin biosynthesis pathway in plants. (**A**): Phenylpropanoid pathway showing the synthesis process of monolignols from phenylalanine, (**B**): monolignol-specific pathway, (**C**): Sub units namely: (guaiacyl), S (syringyl), and H (hydroxyphenyl oxidized from above steps form three-dimensional polymer of lignin.

**Figure 4 ijms-20-04211-f004:**
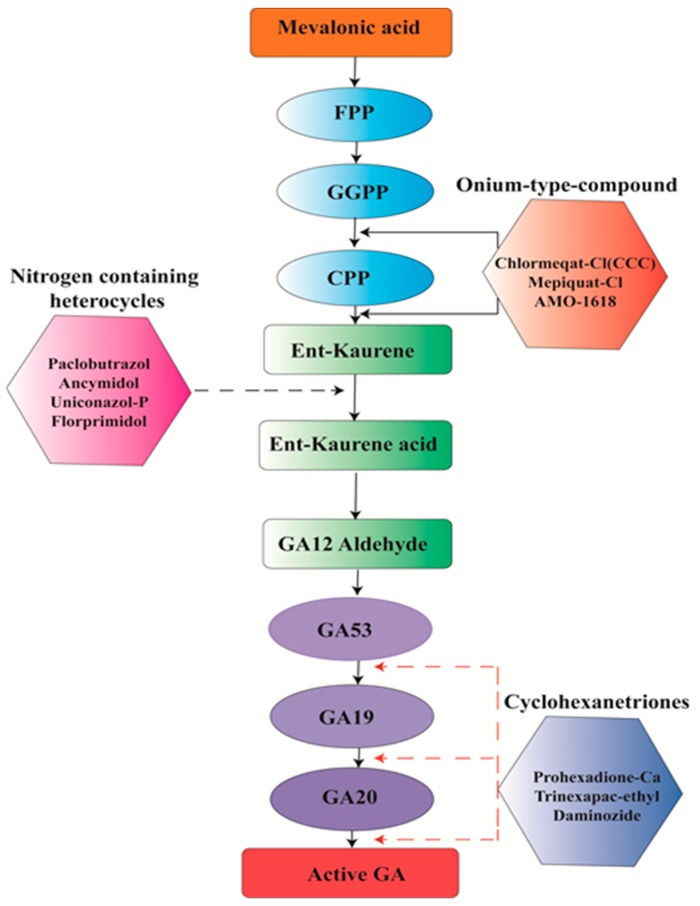
Gibberellic acid biosynthesis with respect to inhibition points by plant growth regulators. Broken line represents minor inhibitor activities. Farnesyl Pyrophosphate (FPP), Geranylgeranyl Pyrophosphate synthase (GGPP), Copalyl Pyrophosphate synthase (CPP).

**Figure 5 ijms-20-04211-f005:**
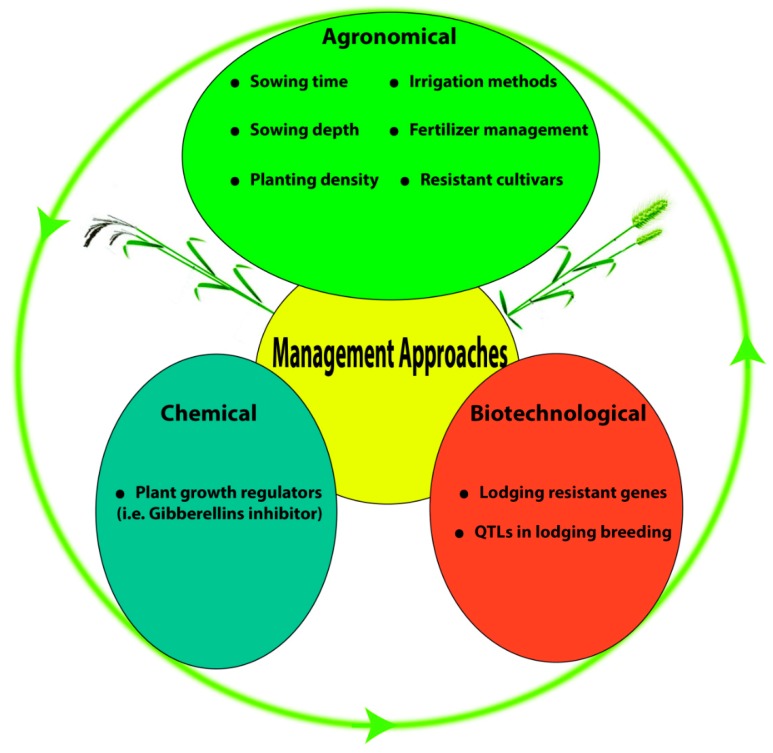
Summary of the most appropriate ways including molecular breeding strategies, chemical control, and agronomical management approaches applying to mitigate field lodging.

**Table 1 ijms-20-04211-t001:** Plant constituents associated with lodging resistance in wheat and rice.

Trait (s)	Crop	Behavior	Reference (s)
**1. Morphological Features**			
Plant height	Wheat and rice	Positively correlated with lodging	[27,29,30,37,39]
Primary inter-nodal length	Wheat and rice	Negatively correlated with lodging resistance	[45]
Culm diameter	Wheat and rice	Strongly positively association with resistance to lodging	[30,36,39,47]
Culm length	Wheat and rice	Positively correlated with lodging	[59]
Culm diameter, panicle weight, panicle length	Rice	Positively correlated with resistance to lodging	[59]
**2. Anatomical Aspects**			
Number of vascular bundles	Rice	Strongly positively associated with resistance to lodging	[63]
Width of mechanical tissue layer	Wheat	Strongly positively associated with resistance to lodging	[37]
**3. Biochemical Features**			
Lignin and cellulose contents	Wheat and rice	Strongly positively correlated with culm strength, secondary cell wall strength, and lodging resistance	[37]
**4-Plant Root System**			
Root lodging resistance	Wheat and rice	Associated to diameter of the root soil cone, bending strength of the crown, and the shear strength of the soil	[70,72,79]

**Table 2 ijms-20-04211-t002:** Quantitative trait loci (QTLs) for plant height and culm thickness in wheat and rice.

Chromosomes/(QTLS)	Flanking Marker Loci	Impact on Plant Traits	Reference (s)
Wheat			
1BS, 4AS, 7BL	Xpsr949-Xgwm18, Xgwm397-Xglk315, Xpsr927-Xpsr350	Shorter plant height	[131]
2AS, 3AS, 5AL	Xpsr958-Xpsr566c, Xpsr598-Xpsr570, Xpsr918b-Xpsr1201a	Culm wall thickness	[131]
4B, 4D	Xgmti538-Xgwm6, XgKm60S-Xgdml29	Shorter plant height	[132]
3A*(QSs-3A),* 3B *(QSs-3B)*	Xwmc527-Xwmc21, Xgwm108-Xwmc291	Stem strength	[133]
*QSd-3B*	Xgwm108-Xwmc291	Culm diameter	[133]
*1A(QPd-1A),2D (QPd-2D)*	Xgwm135-Xwmc84, Xgwm311-Xgwm301	Pith diameter	[133]
*QCwt-2D*	Xgwm311-Xwmc301	Culm wall thickness	[133]
3BL	GWM247 and GWM340	Lignin contents	[37]
1D, 2A, 3A, 4A, 6A, 7D	Xgwm642, Xgdm93, Xgdm93, Xwmc313, Xgwm570, wPt-9690, Xbarc184, respectively	Control Plant height	[30]
3A, 4D	Xwmc264, Xwmc48 8, respectively	Meaningful effect on lodging risk	[30]
3A	Xgwm369	Association with lignin and cellulose	[30]
1A, 6A	Xcfa2153, Xwmc32	Association with gravity of the stem	[30]
3BS	Xbarc102	Reduce plant height	[134]
6A	Xwmc256*-*Xbarc103	Reduce plant height	[137]
*Rice*			
Chr. 1 *(SCM1)*	RM8111-RM8067	Culm strength	[52,128]
*Chr. 6 (SCM2)*	RM6395 and RM5509	Culm strength, increased spikelet number, and grain yield	[52,128,129,138]
Chr. 3 *(SCM3)*	RM15761-RM15782	Culm strength and spikelet number	[52,128,139,140]
Chr. 2 *(SCM4)*	RM3703-RM2468	Culm lodging resistance	[52,128,140]
Chr. 4, 5 *(prl5)* 6,11*(prl11),* 12	C946, C1081, R2171, C82, G1406	Providing resistance to the basal portion of the stalk	[140]
Chr.1, 3, 6, 7, 8, 12	C885, C595, C358, C451, C10122, R3375	Culm diameter	[140]
Chr. 5, 5, 6	R1838, C246, R2549	Lodging resistance from typhoon	[140]
*1(qCL1.1); 2(qCL2.1); 7(qCL7.1)*	id1021344, id2004861, id7002801	Culm length	[59]
*1(qCD1.1); 2(qCD2.1); 7(qCD7.1)*	id1003559, id2007818, id7001246	Culm diameter	[59]
*1(qCS1.1); 2(qCS2.1); 2(qCS2.2); 2(qCS2.3); 6(qCS6.1); 6(qCS6.2)*	id1003559, id2007818, id2006621, id2008112, id6001960, id6010515	Culm strength	[59]

**Table 3 ijms-20-04211-t003:** Resistance genes associated with lodging resistance in wheat and rice.

Gene (s)	Crop	Impact on Plant Traits	Reference (s)
*Rht1 and Rht2*	Wheat	Reduced internode length, Reduce plant height (Wheat)	[145,146,147,148,149]
*Rht3*	Wheat	Reduce plant height (Wheat)	[154]
*Rht5 and Ppd-D1*	Wheat	Reduce plant height	[28,155,156,157]
*Rht7 and Rh8*	Wheat	Reduce plant height	[158,159]
*Rht-B1, Rht-B1a, Rht-B1a, Rht-D1, Rht8, Ppd-B1* and *Ppd-D1*	Wheat	Reduce plant height	[160]
*Rht4, Rht5, Rht8, Rht9, Rht12* and *Rht13*	Wheat	Shortening plant height	[134]
*Rht8* and *Rht-B1b or Rht-D1b*	Wheat	Increased roots and reduced plant height	[163,164]
*Rht14, Rht15, Rht16, Rht18* and *Rht19*	Wheat	Reduced plant height	[166,167]
*Rht22, Rht-B1IC12196* and *Rht-B1f*	Wheat	Dwarfing genes	[166,167,169]
*Rht24*	Wheat	Dwarfing genes	[137]
*COMT (W-cm5-1)*	Wheat	Expressed in root, plant stalk, and leaf tissues	[170]
*semidwarf1 (sd1)*	Rice	Reduce plant height	[19]
*OsCesA1, OsCesA3* and *OsCesA8*	Rice	Cellulose synthases	[174]
*OsCesA4, OsCesA9,* and *OsCesA7*	Rice	Expressed in secondary cell walls	[175]
*OsCLF*, *OsEMF2b* and *OsFIE2*	Rice	Dwarf phenotype	[176]
*OsEMF2b*	Rice	Dwarf phenotype	[178]
*TaCM,*	Wheat	Relationship with the accumulation of stem lignin and stalk lodging mechanical strength	[95]
*PAL, CCR, CCoAOMT, COMT* and *CAD*	Wheat	Lignin biosynthesis enzymes	[180]

**Table 4 ijms-20-04211-t004:** Wheat and rice genotypes along with lodging vulnerability status.

Genotype (s)	Lodging Rating	Reference (s)
**Wheat**		
PBW 343, UP 2338	Tolerant	[249]
Baviacora 92, Seri 82, Star, Munia/Kauz, Weaver	Tolerant	[249]
Baviacora 92	Moderately resistant	[250]
Zhoumai 18, Zhoumai 22 and Pingan 6, Aikang 58	Resistant	[251]
Kalyansona and Sonalika	Resistant	[252]
Savannah, Rialto, Buster, Hereward	Resistant	[47]
Norin 10	Resistant	[253,254]
Hereward, Spark, Cadenza, Mercia	Resistant	[222]
H4564 and C6001	Resistant	[170]
Oslo, Sapphire, AC Foremost, ND695 and AC Vista and Kohika	Tolerant	[27]
**Rice**		
Wuyunjing 23	Resistant	[38,46]
IR-8	Resistant	[255]
Pusa Basmati-1, Pusa-1121, RH-10	Resistant	[256]
Dontokoi, IR24	Resistant	[257]
T5105	Resistant	[258]
Yliangyou2	Resistant	[15]
Takanari	Resistant	[129]
Peiai64, Zhefu802, Liantangzhao, 76-27B, Chunjiang025, Xiushui04, Xiushui63, Jia02-43, ZH222, Bing02-133, Taihunuo, HanfenB, Kinmaze, HZ0302, Jia02-5, Minghui63, Teqing, 9308, Guangsi, K17B, QingreB, 486B, 5N-76B, Bing02-09, DiguB, V20B, Aijao-Nante, Zhaiyeqing8,GuangB, Guichao, lemont, M202, M201, 98–110, Bing02-105, R0308, Shirasenbon, Fukuhibiki	High-yielding semi-dwarf rice cultivars contain *Sd1* gene	[171]

**Table 5 ijms-20-04211-t005:** Role of plant growth regulators for reducing lodging stress in wheat and rice.

Crop	Application Time	Height Reduction	Grain Yield	Reference (s)
**Effect of Chlormequat**				
Wheat	leaf growth stages (from stage 3 to 5)	3.3–14.5%	No effect	[268]
Spring wheat	Second node detectable/third node visible	1–29%	No effect	[269]
Winter wheat	Pseudo stem erection-first node visible	6%	No effect	[270]
Winter wheat	Pseudo stem erection/first node visible	3–15%	No effect	[7]
Spring wheat	Vegetative development	20 cm	Smaller effect	[271]
**Effect of Ethephon**				
Spring wheat		12 cm	No effect	[271]
Spring wheat	Booting stage	1–12%	Increase grain yield (5.4%)	[274]
Wheat	Before anthesis	9.0%	Reduced grain yield (8.3%)	[249]
**Effect of Trinexapac-ethyl**				
**Wheat**	Stem elongation	Reduced plant height	Reduced wheat yield	[279,280]
**Paclobutrazol**				
Wheat	Stem elongation stage	10–20%	Increased grain yield	[5]
Rice	First growing seedlings	20–30%	Increased grain yield	[284]
Rice	15–10 days before heading	15–25%	Increased yield (15%)	[285]
**Prohexadione-calcium (Pro-Ca) and trinexapac-ethyl (TNE) Effect**				
Rice	5–10 days before heading	5–10%	No effects	[287]

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
