# Peer review of "Improving Lodging Resistance: Using Wheat and Rice as Classical Examples"

_ijms, 2019, doi:10.3390/ijms20174211_

Round 1

Reviewer 1 Report

The review paper focuses on the Lodging disorder that mainly occurs in wheat and rice. In my opinion the authors have to talk about tolerance instead of resistance. This term is not appropriate form biological and agronomic point of view. 

The review should be improved on physiological and biochemical pathway involved in the traits that prevent or increse the sensitivity to lodging. There is a good overview on the morphological, genetic and agronomic aspects correlated to lodging. 

The review should be improved including the discussion on:

- plant hormones and lodging at roots and shoot relationship. The ethylene should be considered with a deep level of discussion. The cross-talk among plant hormones should be considered even if there are not many papers focused on this Research topic. 

- Root architecture, functionality, and aerenchima formation. These factors should be linked to lodging incidence. 

-the authors should be critical and provide their opinion among the published papers. Reporting what is contradictory or weak and what need to be clarified with further investigations. 

Author Response

Dear Reviewer,

We are very thank you for your thoughtful and supportive comments. We have carefully considered each comment and extensively revised the manuscript accordingly. The following are answers and explanation with respect to your comments.

Thank you again for your consideration

Regards,

Prof. Hongqi Si, corresponding author

Reply to Reviewer # 1:

Comments and Suggestions for Authors

Point 1: The review paper focuses on the Lodging disorder that mainly occurs in wheat and rice. In my opinion the authors have to talk about tolerance instead of resistance. This term is not appropriate form biological and agronomic point of view.

Response 1: Thank you for your comment. According to discussed traits that we mentioned in our submitted, the most suitable word is lodging resistance rather than tolerance, which was used by many researchers earlier in their published articles. For instance “Lodging resistance of winter wheat (Triticum aestivum L.): Lignin accumulation and its related enzymes activities due to the application of paclobutrazol or gibberellin acid” written by Peng et al. 2014, secondly, lodging resistance in cereals by PM Berry - Crop Science, 2019, and article exploring the traits for lodging tolerance in wheat genotypes: a review R Khobra et al, 2019. We also used the term tolerance wherever it was appropriated in the article. 

Point 2: The review should be improved on physiological and biochemical pathway involved in the traits that prevent or increase the sensitivity to lodging. There is a good overview on the morphological, genetic and agronomic aspects correlated to lodging.

Response 2: Thank you for your suggestion. We have included the lignin pathway in the revised manuscript.

Point 3: Plant hormones, lodging at roots and shoots relationships. The ethylene should be considered with a deep level of discussion. The cross-talk among plant hormones should be considered even if there are not many papers focused on this Research topic.

Response 3: Thank you for your comment. We have added a subchapter about ethylene and hormones role in wheat and rice that have a role in lodging in the revised manuscript.

Point 4: Root architecture, function, and aerenchyma formation. These factors should be linked to lodging incidence.

Response 4: Thank you for your comment. We have revised the manuscript according to your suggestions and discussed root architecture, functionality, and aerenchima formation in the revised manuscript.

Point 5: The authors should be critical and provide their opinion among the published papers. Reporting what is contradictory or weak and what need to be clarified with further investigations.

Response 5: We have raised some questions that still remained to be answer related to lodging resistance and still need further investigation to answer unsolved questions for getting information to understand complex lodging trait.

Reviewer 2 Report

Review report

Improving Lodging Resistance:  Using Wheat and 2 Rice as Classical Examples

Figure 1: Length spelling, start with capital letters: Width, Small, Head and Internode

What is the use of ‘logded plant’ in figure? What are the factors causing lodging?

Line 188: the?

Table 1: Anatomical traits: and Biochemical features: Associated does not mean anything? Positive or negative? Be clear. 4-plant root??

Table 3: Impact on plant traits missing for genes?

Line 286, 359: do not start with a reference number.

Line 291: with Koshihikari.

Line 314: Gene’s Association???

Line 395: Plant height reduction???

Line 410-411: Grammar correction needed.

Line 421: the first

Line 467: ‘soil was high moisture’ – bad phrase!

Line 479: “Ma et al. [186] stated that appropriate regulated deficit irrigation”- bad grammar! What do you mean by appropriately regulated?

Line 499: NH+ application???

Line 504: Kong, Sun, Wang, Liu, Feng, Si, Zhang, Li and Li??? Do you mean Kong et al.

Line 510: reduced the root penetration

Line 719: planned

Line 737: morphological what?

Line 743: dire need?

Author Response

Dear Reviewer,

We are highly thankful to you for your valuable comments to improve the manuscript. We have carefully considered each comment and extensively revised the manuscript accordingly.

Thank you again for your consideration.

Regards,

Prof. Hongqi Si, corresponding author

 Reply to Reviewer # 2:

 Comments and Suggestions for Authors

Figure 1: Length spelling, start with capital letters: Width, Small, Head and Internode What is the use of ‘logded plant’ in figure? What are the factors causing lodging? Line 188: the? Table 1: Anatomical traits: and Biochemical features: Associated does not mean anything? Positive or negative? Be clear. 4- plant root ?? Table 3: Impact on plant traits missing for genes? Line 286, 359: do not start with a reference number. Line 291: with Koshihikari. Line 314: Gene’s Association??? Line 395: Plant height reduction??? Line 410-411: Grammar correction needed. Line 421: the first Line 467: ‘soil was high moisture’ – bad phrase! Line 479: “Ma et al. [186] stated that appropriate regulated deficit irrigation”- bad grammar! What do you mean by appropriately regulated? Line 499: NH+ application??? Line 504: Kong, Sun, Wang, Liu, Feng, Si, Zhang, Li and Li??? Do you mean Kong et al. Line 510: reduced the root penetration Line 719: planned Line 737: morphological what? Line 743: dire need?

Response: Thank you for your comments. We thoroughly edited the whole manuscript and improved the grammar and English per your suggestion as well made the corrections you highlighted in your comments. 

Round 2

Reviewer 1 Report

the manuscript has been substantially improved and answers were satisfactory. Therefore, in my opinion it can be accepted for publication in the journal.